# Implementation of Routine Endoscopy with Narrow Band Imaging in the Evaluation of Oral and Upper Airways Lesions in Oral Chronic Graft-Versus-Host Disease: A Preliminary Study

**DOI:** 10.3390/jpm12101628

**Published:** 2022-10-01

**Authors:** Letizia Nitro, Carlotta Pipolo, Paolo Castellarin, Andrea Sardella, Antonio Mario Bulfamante, Beatrice De Marco, Gabriele Magliano, Giovanni Grillo, Giovanni Felisati, Alberto Maria Saibene

**Affiliations:** 1Interdisciplinary Center for Oropharyngeal Pathology (CIPO), Otolaryngology Unit, Santi Paolo e Carlo Hospital, Department of Health Sciences, Università Degli Studi di Milano, 20146 Milan, Italy; 2Department of Biomedical, Surgical and Dental Sciences, Odontostomatology Unit, ASST Santi Paolo e Carlo, Università degli Studi di Milano, 20142 Milan, Italy; 3Bone Marrow Transplantation Unit, ASST Grande Ospedale Metropolitano Niguarda, 20162 Milano, Italy

**Keywords:** graft-versus-host disease, hematopoietic stem cell transplantation, endoscopy, narrow-band imaging, oral squamous cell carcinoma, intrapapillary capillary loops, oral potentially malignant disorders

## Abstract

(1) Background: The aim of our study is to investigate the main oral lesion patterns in patients with oral graft-versus-host disease and to describe and validate the use of endoscopy enhanced with narrow-band imaging (NBI) as a personalized, reliable and user-friendly tool for the early detection of oral potentially diseases. (2) Methods: We retrospectively evaluated the medical records of 20 patients with chronic GVHD and with oral manifestations, who were referred to our “Interdisciplinary Center for Oropharyngeal Pathology (CIPO)” from January 2017 to July 2022. (3) Results: Data on GVHD, oral localization and NBI endoscopic evaluation are collected. A total of six mucositis, one mucosal erythematous change, ten lichenoid-like changes, eight erosive lesions, one leukoplakia, two erythroplakia and two case of blisters were observed. Two vascular abnormalities were seen with NBI, leading to one excisional biopsy. The patient was diagnosed with squamous cell carcinoma. (4) Conclusion: Our study is the first to highlight the relevance of the routine use of endoscopy with NBI in patients with oral chronic GVHD. We highlighted its role as a reliable, reproducible, easy-to-use and tailor-made tool in the follow-up of those patients and to allow an earlier identification of aberrant neoangiogenesis related to oral potentially malignant disorders and oral cancer.

## 1. Introduction

Graft-versus-host disease (GVHD) is one of the most severe complications of hematopoietic stem cell transplantation (HSCT) and it represents a major limit to its wide application [1,2]. GVHD classification is based on two major phenotypes: acute (aGVHD) and chronic (cGVHD) [1]. aGVHD typically shows an early onset with rapid and dramatic evolution, dominated by an inflammatory pattern which inevitably leads to a multi-organ failure (in particular liver, lungs, bone marrow, skin, and gut involvement) with a poor prognosis [3,4]. cGVHD has a high prevalence in post-transplanted adults (almost 50%) with a late onset, usually about after 100 days post-HSCT, and displays more autoimmune and fibrotic features, such as cutaneous and mucosal lichenoid-like changes [1,4,5]. cGVHD mainly affects oral and oropharyngeal mucosa, salivary and lacrimal glands, and subcutaneous connective tissues [4]. The improvement of available tools for GVHD prevention led to an increase in patients eligible for stem cell transplants [1,2,6]. Despite these factors, GVHD prevalence has remained constant and even increased in its chronic form [1,6]. This is justified by the introduction of advanced therapies that have allowed transplants to be performed even in cases of partial compatibility between donor and recipient [1,6]. The histocompatibility of the transplanted cells is a key point of the pathogenesis of GVHD, and it could also related to the severity of the clinical manifestations [5]. This pathological process is due to a lack of recognition between the human leukocyte antigens (HLA) of the major histocompatibility complex (MHC) expressed on the leukocytes’ cell surface [1]. The cGVHD cellular reaction is widely identified as a T-Helper 2 (Th2) predominant with the activation of a Th2 fibrogenic cytokines cascade, such as IL-2, IL-10, and TGFβ [4]. The activation of macrophages and fibroblasts, dominated by the action of the cytokine-apoptosis-transforming growth factor-beta (TGFβ) and platelet-derived growth factor-mediated pathways (PDGF), leads to fibrotic tissue changes with inevitable alterations of the affected host sites [4]. In addition, the dysregulation of B-cells, due to low levels of regulatory T-cell (Treg), is demonstrated to be the reason for the findings of autoreactive antibodies in cGVHD [4,7]. The pathogenesis of oral cGVHD seems to be related to specific single-nucleotide polymorphism (SNP) of FCGR2A in donors. It allows us to stratify the risk of developing the oral disease [5,8]. Furthermore, the specifical salivary proinflammatory cytokine interleukin (IL)-6 (IL-6) and IL-1alpha (IL-1α) are commonly identified in the oral cavity of those patients and their concentrations seem to be related to the severity of the disease [5,9]. The diagnosis of oral GVHD is mainly based on clinical assessment of oral lesions, which are described as “*lichen planus-like changes*”; erosive, ulcerative, and erythematous lesions are not considered univocal diagnostic for oral GVHD [5,9,10]. 

From a symptomatic point of view, patients with oral GVHD suffer from recalcitrant mucositis, increased risk of dental caries, xerostomia, taste alterations, feeding impairment, and relapsing infection [9,10,11,12]. One of the most critical issues in the follow-up of oral GVHD is represented by the lack of assessment of a reliable tool for the routinary evaluation of oral lesions, also in relation to the well-known risk of developing squamous cell carcinoma of the oral cavity and oropharynx [13,14]. A solution to this issue can be offered by narrow-band imaging (NBI). This technology is based on the use of specific optic filters that isolate two wavebands, the green and blue ones, that share absorption peaks with hemoglobin (540 nm and 415 nm) from a cold light source and leads to the enhancement of mucosal and submucosal vascularization with the identification of aberrant blood vessel or vascular patterns, possible expression of angiogenesis [15,16]. In comparison to oral biopsy, it has less site morbidity and it is less invasive for the patients [16,17]. NBI could also be a useful tool to guide the biopsy in case of small or difficult-to-reach lesions. According to Takano et al., NBI enhances the vascular architecture and the identification of the intra papillary capillary loops (IPCL) and their IPCL classification describes four types of vascular patterns: type I and II are typical of non-neoplastic lesions and type III and IV, are, respectively, distinctive of oral potentially malignant disorders and cancer [18]. In type III, the increase in length and dilatation of blood vessels can be observed, up to type IV with the typical destruction of the loops at the terminal branches and angiogenesis [18]. The use of conventional WLI endoscopy combined with NBI in the otolaryngological field is widely acknowledged, particularly for the diagnosis of early-stage oral cavity/oropharynx/laryngeal SCC [19,20,21] but also for the follow-up of oral chronic inflammatory lesions with a potential of malignancy, such as Lichen Planus [22,23]. Other advantages of this technique are that it is easily adaptable to the patient’s clinical needs, it is easy to perform, it allows an easy repetition of the exam, it has short learning curves and it led to the possibility of saving images and intraoperative use in case of biopsy or for the analysis of the resection margins [24,25,26].

The purpose of our study is to investigate the main oral lesion patterns in patients with oral GVHD referred to our outpatient clinic, describe the key role of the routine multidisciplinary otolaryngological and odontoiatric examination, and validate the role of endoscopy with NBI in the evaluation of the oral cavity and oropharynx as a user-friendly tool and replicable tool in the evaluation of the oral cavity and oropharynx, and facilitating early detection of oral potentially malignant disorders and/or squamous cell carcinoma. 

## 2. Materials and Methods

This study was designed as a retrospective review. Due to its retrospective nature, it was granted exemption from the Internal Review Board of the San Paolo Hospital, Milan. We reviewed medical charts and endoscopic recordings of patients aged 18 or older with oral manifestations of cGVHD following hematopoietic stem cells transplant accessing our third-level multidisciplinary outpatient clinic for oro-pharyngeal disease. aGVHD and other known concomitant autoimmune oral conditions were considered exclusion criteria. Records between January 2017 and July 2022 were reviewed.

For all eligible patients, we recorded gender, age, anamnesis, main information about HSCT and GVHD and full description of oral features. Oral endoscopy and upper airways endoscopy recordings were reviewed by two authors (LN and AMS) to describe white light imaging (WLI) and NBI appearance of oral manifestations of cGVHD. 

### Study Population and Patients Management

From January 2017 to July 2022, 20 patients consecutively referred to our “Interdisciplinary Center for Oropharyngeal Pathology (CIPO)” with a diagnosis of chronic GVHD with oral involvement. All these patients were considered eligible according to our inclusion and exclusion criteria and their medical records were retrospectively reviewed. All patients enrolled in the study underwent a complete ENT evaluation and an oral and upper airways WLI endoscopy performed was combined with the use of NBI (OLYMPUS, ENF version 3). This allowed the recognition of different features of GVHD and early detection of any malignant and premalignant oral cavity lesions. Data were collected and elaborated in terms of purely descriptive statistics with Excel (Microsoft Corporation, Redmond, WA, USA). 

## 3. Results

We collected the data of 20 patients with oral cGVHD, 9 females and 11 males with an average age of 52.61 ± 13.81 years. Nineteen patients were transplanted at the Bone Marrow Transplant Center of Niguarda Hospital in Milan, and only one patient was transplanted in a different Bone Marrow Transplantation Unit. The study period covered over 5 years and 7 months (from 1 January 2017 to 31 July 2022), with an average follow-up time for each patient of 12.4 ± 18.47 months. Demographic and clinical data are collected and summarized in Table 1. 

All enrolled patients were transplanted with hematopoietic stem cells in a period between 11 and 276 months before, with an average age at allogeneic-HSTC of 51.1 ± 14 years.

The diagnoses were: chronic lymphocytic leukemia (CCL) in one patient, acute lymphoblastic leukemia (ALL) in four patients, acute myeloid leukemia (AML) in nine cases, chronic myeloid leukemia (CML) in one patient, myelofibrosis developed from polycythemia vera in one patient, Hodgkin’s lymphoma in two patients, non-Hodgkin’s lymphoma in one patient and plasma cell leukemia (PCL) in one patient. In four cases it was possible to use a matched related donor’s stem cells and in two cases a haploidentical donor was chosen. Matched unrelated donor (MUD) cells were selected in 16 patients. Seven of the patients referred have also previously developed an acute GVHD. Table 2 summarizes all the information about allogenic-HSCT.

All patients showed oral localization of chronic GVHD and the median time between HSCT and the development of oral GVHD was 11 months. During the first ENT evaluation, all the patients were orally symptomatic; the most frequent reports were pain (*n* = 20), oral burning sensation (*n* = 9), feeding impairment (*n* = 2), recalcitrant oral infection (*n* = 4), and caries (*n* = 8). All patients were subject to the visual analogue scale (VAS) for the establishment of the burden of oral disease with a median value of 7.5.

A full ENT examination and oral and upper airway endoscopy were performed with the use of NBI (OLYMPUS, ENF version 3). The results of the primary endoscopic examination of the oral cavity and oropharynx are summarized in Table 3.

The main involved sites at the first ENT evaluation were, respectively, buccal mucosa (*n* = 20), hard and soft palate (*n* = 7), oropharynx (*n* = 4), ventral portion of the tongue (*n* = 5), gingival mucosa (*n* = 2), lips (*n* = 2) and vestibular mucosa (*n* = 3). Involvement of the tongue base was detected in two patients. None of the patients showed laryngeal or hypopharyngeal lesions. 

A WLI oral endoscopy was routinely used for the clinical examination of patients together with the evaluation of the oral cavity and oropharynx through the use of the NBI technique. Several patterns of oral and oropharyngeal lesions were observed such as mucositis (*n* = 6), mucosal erythematous changes (*n* = 1), lichenoid-like lesions (*n* = 10), and erosive lesions (*n* = 8), leukoplakia (*n* = 1), erythroplakia (*n* = 2), and blisters (*n* = 2). In our patient group, vascular abnormalities or intrapapillary capillary loops were seen in three patients (Figure 1, Figure 2, Figure 3, Figure 4, Figure 5, Figure 6 and Figure 7). The first one (patient n°3, Table 1, Table 2 and Table 3) had a squamous cell carcinoma of the lower lip after 4 years from HSTC before the first evaluation in our Interdisciplinary Center for Oropharyngeal Pathology (CIPO) and underwent a biopsy of the tongue right margin in June 2022 because of the persistence of leukoplakia with evidence of IPCL in the surrounding mucosa with a diagnosis of epithelial verrucous hyperplasia. The second one (patient n°2, Table 1, Table 2 and Table 3) has two previous tongue SCC and one tonsillar SCC, respectively, 20 and 21 years after HSCT. Due to his anamnesis, he underwent an excisional biopsy of the suspect area at the NBI located at the left amygdalo-glossus sulcus, that revealed the presence of squamous malignant cells. An MRI with contrast enhancement was performed with evidence of invasion of the left masticatory space and muscles, oral pelvis with an invasion of the left mylohyoid and genioglossus muscles, and with extension at the left epiglottic vallecula. 

In another patient (patient n°9, Table 1, Table 2 and Table 3) with a history of lung and colon adenocarcinoma after HSCT, a leuko-erythroplakia of the left buccal mucosa with II type of IPCL was found during oral evaluation. A positron emission tomography/computed tomography (PET/CT) performed in June 2022 for carcinoembryonic antigen (CEA) increase revealed no pathologic absorption in the head and neck district, nonetheless, a close follow-up is performed (every 2 months).

## 4. Discussion

### 4.1. How to Manage and Stage GVHD?

The management of cGVHD is still a challenging issue today, with a prevalence of about 50% in patients who have received HSCT [5]. According to Mays et al., the burden of cGVHD and the incidence of oral cGVHD are constantly growing [27]. Among the major factors for the development of cGVHD are, reportedly, a gender mismatch between donors and recipients, a major allelic mismatch between HLA of the donor and the recipient, TBI conditioning, unrelated donors, older recipient age, and prior acute GVHD [7,27,28,29]. Chronic GVHD could affect different sites with typical tissue changing, hence the feature of lichen planus-like changes in the mucosa was introduced [30,31]. Despite the fact that oral and oropharyngeal localizations of cGVHD are well known and described in the literature [1,4,5,9,11,27,32] what is less investigated is the significant impact on the quality of life of those patients, mainly due to the burdensome symptoms [9,10,11,12,27,33]. To the best of our knowledge, only two tools were previously used for the assessment of the severity of oral symptoms, the visual analog scale (VAS) and health-related quality of life (HRQL) in the FACTG version 4 [9,34,35]. In clinical practice, a real issue raised by the study of Fall-Dickinson et al. [5] is represented by the diagnosis and classification of oral cGVHD. In fact, using the currently available options, the diagnosis could only be based on clinical criteria, such as the NIH oral cGVHD clinical scoring instrument, which is based on a “three-grade” classification (erythema, lichen-like changes, and ulcerations) [36] or the proposal from the study of Triester et al. of the use of intraoral photographs as a reliable tool [37]. The oral mucositis rating scale (OMRS) by Schubert et al. has been suggested as a new diagnostic tool based both on the anatomic site of oral lesions (lips, labial mucosa, right and left buccal mucosa, dorsal-lateral-ventral tongue, oral pelvis, hard and soft palate, and attached gingiva) and on the type of alteration (erythema, atrophy, hyperkeratosis, lichenoid changes, and edema) and severity (rated with a score from 0 to 3 for all the subsites) [38]. 

### 4.2. Therapies Cause Cancer?

The importance of a personalized follow-up of the patients with oral GVHD is also related to the possibility of developing a de novo squamous cell cancer of the head and neck [39]. It is established that in solid-organ transplanted patients there is a 20 times higher risk of developing secondary malignancy, compared to the general population, mainly due to higher dose of immunosuppressive therapies [40,41]. Douglas et al. showed that the most affected site of squamous cell cancer in the head and neck after HSCT was the oral cavity, in particular the tongue, and they showed that cancer in these patients had a more aggressive progression and a poorer prognosis [42]. In our patient sample, all of the patients were treated with a conditioning chemotherapy regimen, and, among them, three patients also underwent radiotherapy. Conditioning regimens before hematopoietic stem cells transplant has two goals: disease reduction and “graft-versus-host” reaction prevention [43]. In the literature, two ways to achieve these results are described: total body irradiation (TBI) in supra lethal doses and chemotherapy with non-overlapping toxicities [43,44]. Chemotherapeutic regimes could be, respectively, classified into Conventional Intensity Conditioning (CIC) or Myeloablative (MAC) and Reduced Intensity Conditioning (RIC) [45,46,47]. The difference is mainly based on the myelosuppressive effect which leads to the immunological defenses of the host being overcome and the eradication of neoplastic cells [43]. Nevertheless, those therapies induce several side effects. MAC conditioning regimens are usually chosen in younger patients (<55 y/o) with no major comorbidities (such as cardiologic, pulmonary, and renal) [33]. On the other hand, a more aggressive conditioning therapy could promote the so-called “graft-versus-tumor reaction” that allows a reduction in the burden of the disease [47]. There is an association between TBI and secondary solid tumors in transplanted patients, but nowadays it is still debated. On the other hand, there is no evidence of a relation between chemotherapeutic regimens and secondary malignancy [43,48]. 

### 4.3. What about Risk Factors?

As mentioned above, the existing risk of the development of squamous cell carcinoma of the head and neck is related to different predisposing factors. It is important to underline that for patients with oral GVHD, different risk factors play a major role compared to those for the general population [49,50]. Male gender, tobacco, regular alcohol consumption, poor oral hygiene, and immune deficiency are the most cited risk factors and predisposing conditions for the development of squamous cell carcinoma of the head and neck [49,51,52]. Additionally, lichen planus as an oral potentially malignant disorder was mentioned as a condition contributing to cancerogenesis, especially in the elderly [49,53]. In our group, only three patients are active smokers, one patient is a former smoker and no patients reported regular alcohol consumption. Another important risk factor is represented by the human papillomavirus infection and its oncogenetic role [52,54]. According to Zhang et al., conditioning therapies led to a reduction in immune surveillance that could be manifested in different ways such as a higher tendency to the growth of cellular genetic abnormalities, acquired defects in the DNA repair process, and easier oncogenic virus infections [55]. In the literature, the role of oral acute GVHD as a predisposing factor, both for oral chronic GVHD and for the development of SCC, still remains unclear [56]. Oral aGVHD seems to show different clinical features, such as papular/nodular/ulcerative changes of the oral mucosa, but in a few patients, there was an “overlap syndrome” in which acute and chronic GVHD could be seen [29,56]. In our population, we reported seven cases of acute GVHD, among which two patients developed oral squamous cell cancer.

### 4.4. Why NBI?

In consideration of all the findings above mentioned, it can be seen that patients with oral chronic GVHD need an adequate and tailor-made follow-up, in particular, to ensure an early-stage detection of a secondary SCC [5,9,10,11,40]. A periodic evaluation of oral lesions and the setting up of appropriate therapeutic regimens for symptom control are mandatory. The need to choose an effective diagnostic method that could lead to rapid identification and characterization of oral lesions, in particular those with potential for malignancy, emerges. As stated by Douglas et al., lesional biopsy should be reserved only for the case of persistent lesions with suspect characteristics, such as bleeding, increasing pain, thickening, and ulceration [40]. A prompt diagnosis is also fundamental for the detection of early-stage squamous cell oral cancer and, furthermore, this could be related to a better prognosis and higher overall survival [40]. In our experience, the use of endoscopy for the evaluation of oral lesions has shown promising results. Every patient in our cohort underwent a full clinical examination combined with oral and upper airway endoscopy with white light imaging (WLI) combined with the use of narrow band imaging (NBI). Although this method has a widely recognized role in the literature, to the best of our knowledge, this is the first study that investigates the use of NBI in graft-versus-host disease. Interestingly, some patients did not show typical disease lichenoid changes of the oral or oropharyngeal mucosa, but, through the use of NBI, those alterations were uncovered, granting the arrangement of a proper follow-up. Moreover, in patient 2, 3 and 9, NBI has been an important support to biopsy, both in the decision and in the targeting process. In our experience, the use of NBI is a reliable and tailor-made tool for the early detection of oral potentially malignant disorders and it also guides the clinical management of patients with oral GVHD, such as the timing of the biopsy for suspect lesions. Moreover, it seems to have a valid relationship with the efficacy of local anti-inflammatory therapies and it could demonstrate a reduction in hyperemia and hyperplasia of the mucosa in the case of mucositis. Another important consideration is the role of the therapeutic alliance between otolaryngologists, oral medicine specialists and hematologists in the management of oral chronic graft-versus-host disease: oral involvement in this disease should never be underestimated because of the potential development of the secondary squamous cell of the head and neck and it is mandatory for those patients to be addressed to a specialistic for a multidisciplinary second look. This could indeed change the natural history of the disease in some patients with oral chronic GVHD and, as a result, increase survival, time free from disease, and improve their quality of life. 

The present retrospective study is based on a small sample, limiting the significance of our results. Another limitation is the follow-up time, still moderate to grant a significant analysis on cancer development or relapse, as on overall patient survival due to squamous cell carcinoma development. Nevertheless, we hope to increase our cohort and experience to secure better results in future studies.

## 5. Conclusions

Our study is the first to highlight the relevance of the routine use of endoscopy with WLI combined with NBI in the multidisciplinary evaluation of patients with oral chronic graft-versus-host disease. The main advantages are represented by the possibility to achieve a reliable, reproducible, easy, and tailor-made tool in the follow-up of those patients and allow earlier identification of aberrant neoangiogenesis related to oral potentially malignant disorders and oral cancer. A limitation of our study is represented by the small sample analyzed; further studies are needed to validate the efficacy of oral and upper airway endoscopy and narrow band imaging in this cohort of patients. 

## Figures and Tables

**Figure 1 jpm-12-01628-f001:**
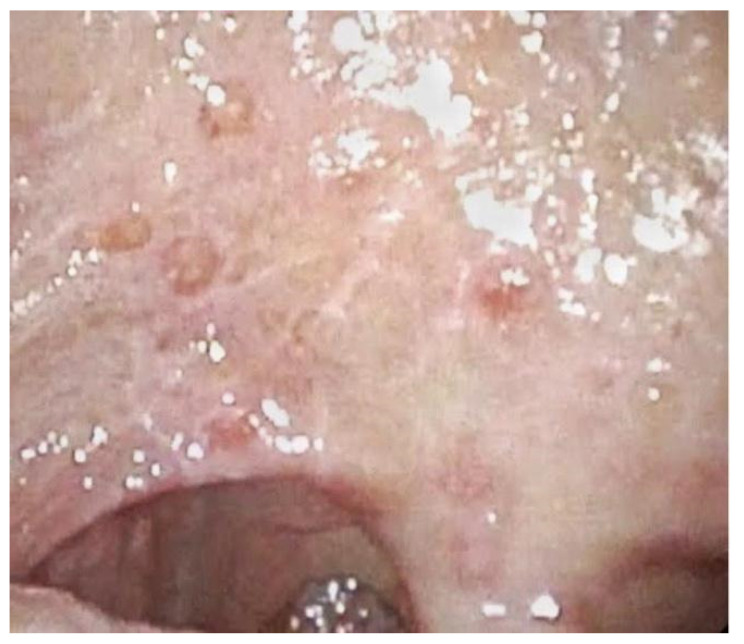
Oral mucositis with blisters.

**Figure 2 jpm-12-01628-f002:**
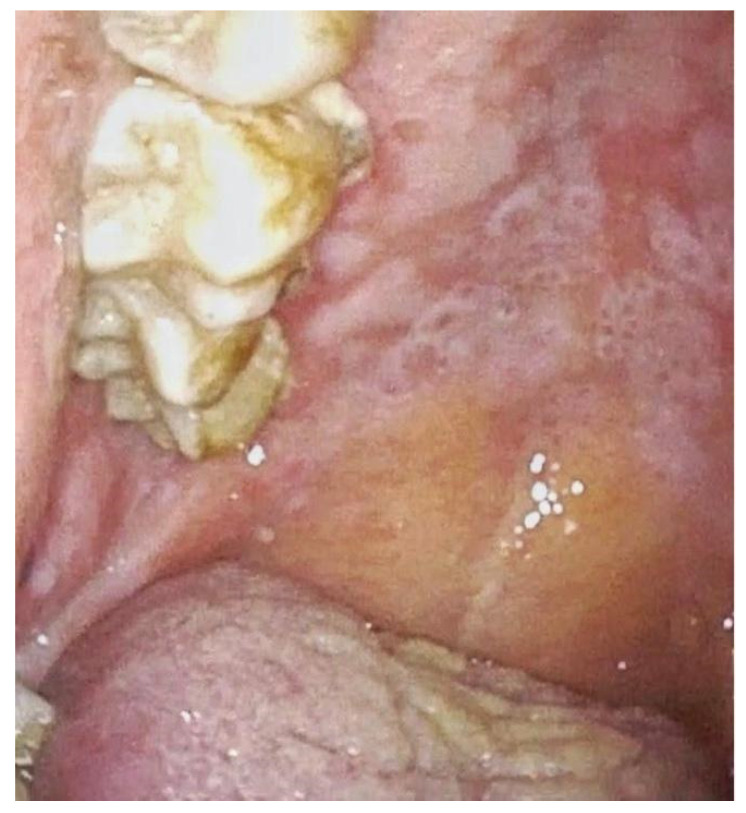
Leucoplakia of the hard palate with erosive spots and erythematous mucosa.

**Figure 3 jpm-12-01628-f003:**
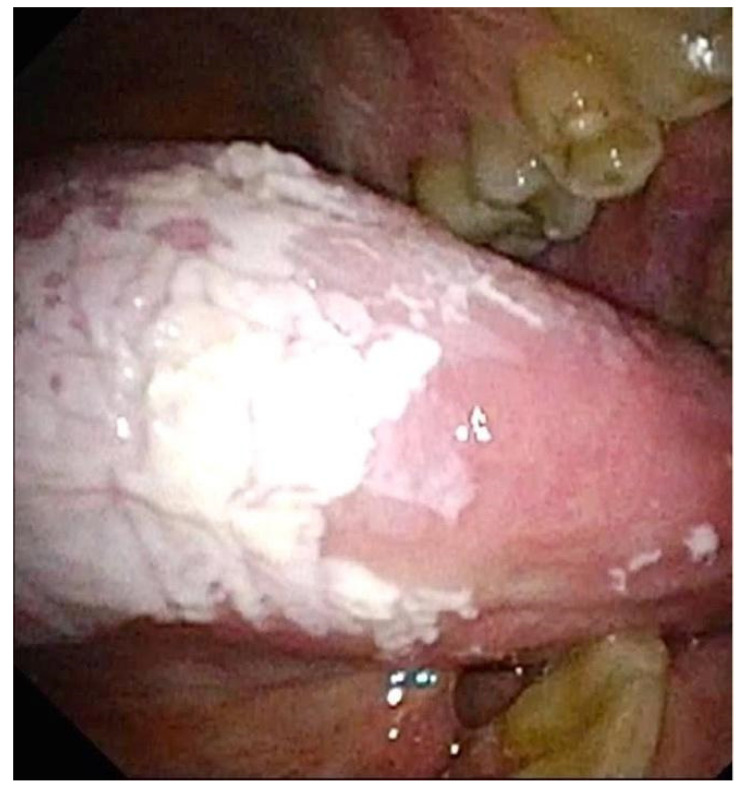
Leucoplakia of the tongue.

**Figure 4 jpm-12-01628-f004:**
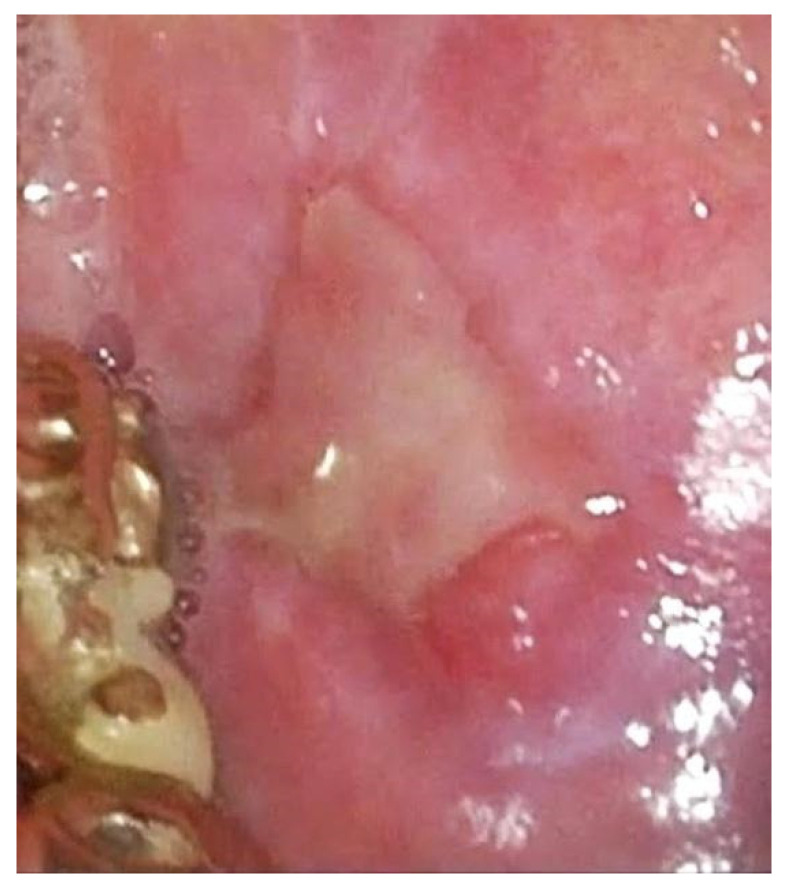
Ulcerated erythroplasic lesion in white light imaging.

**Figure 5 jpm-12-01628-f005:**
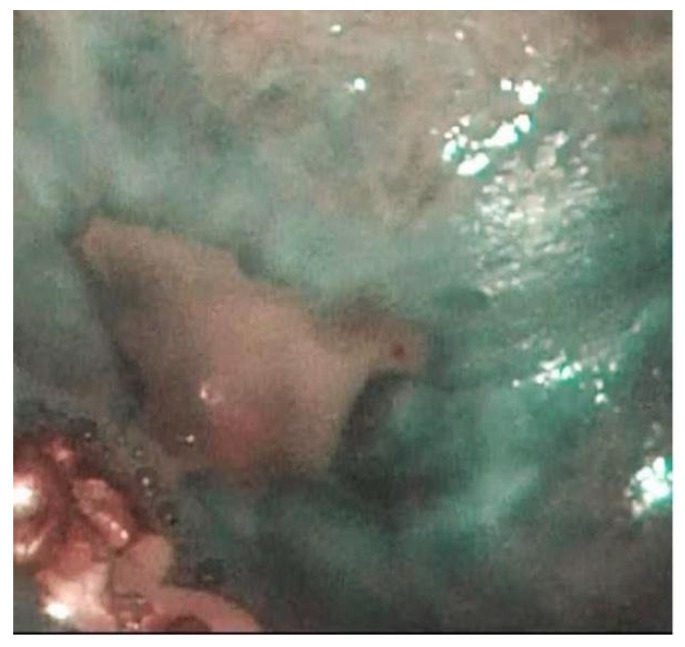
Erythroplasic lesion with peripherical vascular dark spots enhanced with narrow band imaging.

**Figure 6 jpm-12-01628-f006:**
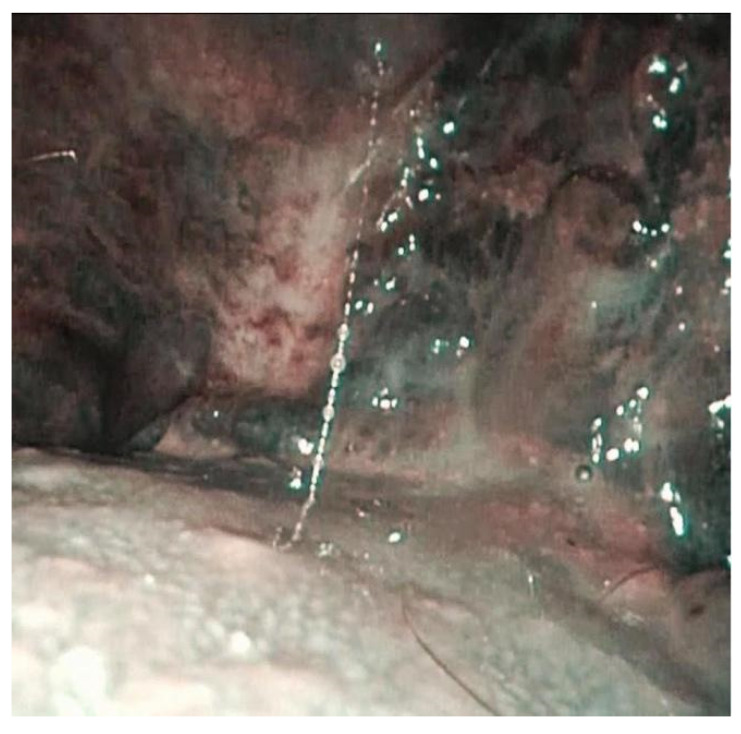
Intrapapillary capillary loop enhanced with narrow band imaging.

**Figure 7 jpm-12-01628-f007:**
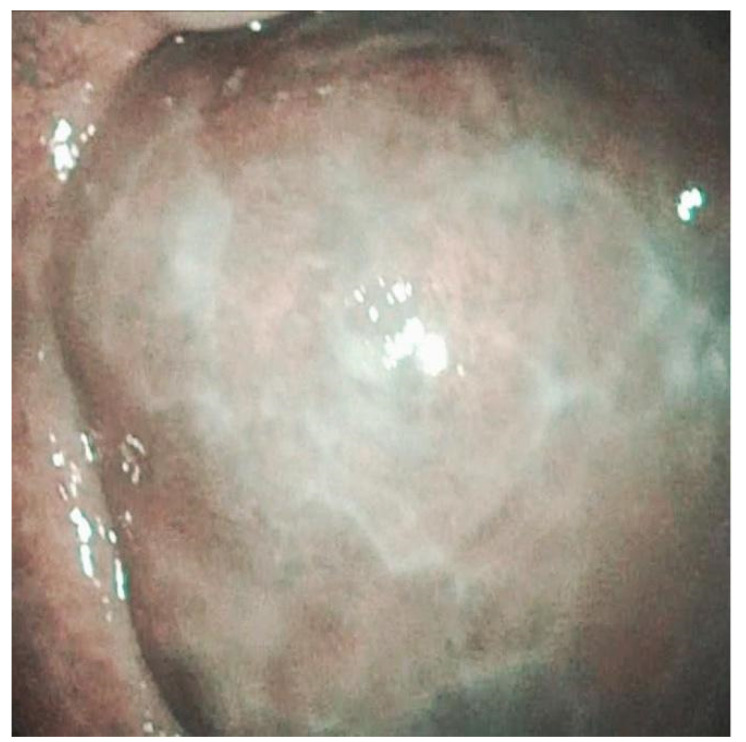
Lichenoid changes of the oral mucosa.

**Table 1 jpm-12-01628-t001:** Demographic and clinical data.

Patient	Sex	Age	Relevant Comorbidities	Alcohol Consumption and Smoking Habits
1	F	67	Gastric resection for gastric cancer; genital recalcitrant herpes zoster virus infection.	Non-smoker, no alcohol consumption.
2	M	39	Tongue cancer in 2017 and 2020, left tonsillar cancer in 2019, follicular thyroid cancer treated with right hemithyroidectomy.	Non-smoker, no alcohol consumption.
3	M	52	Polymyositis; encephalitis in 2018; squamous cells carcinoma of the lower lip in 2018 treated with surgery; Osteoporosis.	Non-smoker, no alcohol consumption.
4	M	49	Chronic gastritis.	Former smoker, no alcohol consumption.
5	F	49	Hashimoto’s thyroiditis; paroxysmal supraventricular tachycardia; cerebral arteriovenous malformation; major depressive disease.	Non-smoker, no alcohol consumption.
6	F	55	Chronic gastritis; hypertension; abdominal aortic aneurysm.	Active smoker, no alcohol consumption.
7	F	67	Hypertension; right upper lobe lung adenocarcinoma in follow-up; osteoporosis; chemotherapy-treated recurrent colon adenocarcinoma	Non-smoker, no alcohol consumption.
8	F	69	Glucose-6-phosphate dehydrogenase (G6PD) deficiency	Non-smoker no alcohol consumption.
9	F	23	No other comorbidities.	Non-smoker, no alcohol consumption.
10	F	72	Hypertension	Active smoker, no alcohol consumption.
11	M	34	No other comorbidities.	Non-smoker, no alcohol consumption.
12	M	50	No other comorbidities.	Active smoker, no alcohol consumption.
13	M	49	Anxious depressive disorder.	Non-smoker, no alcohol consumption.
14	M	39	Low grade glioma treated with surgery in 2005.	Non-smoker, no alcohol consumption.
15	M	62	Hypertension	Non-smoker, no alcohol consumption.
16	M	67	No other comorbidities.	Non-smoker, no alcohol consumption.
17	F	41	Bilateral sensorineural hearing loss induced by chemotherapy drugs	Non-smoker, no alcohol consumption.
18	F	63	Acute kidney failure in 1970; Hashimoto’s thyroiditis; Major depressive disorder; chronic intestinal polyposis.	Non-smoker, no alcohol consumption.
19	F	31	No other comorbidities.	Non-smoker, no alcohol consumption.
20	M	44	No other comorbidities.	Non-smoker, no alcohol consumption.

**Table 2 jpm-12-01628-t002:** Hematopoietic stem cell transplant (HSTC) recordings.

**Diagnosis**	CLL	1
	ALL	4
	CML	9
	MFI	1
	HL	1
	NHL	2
	PCL	1
	AML	1
**Type of Stem Cell Donor**	MUD	16
	Haploidentical	2
	Related	2
**Conditioning Therapy**	ChT RIC	10
	ChT MAC	7
	ChT + RT MAC	3
**Acute GVHD**	Yes	7
	No	13
**Chronic Oral GVHD**	Yes	20
	No	/
**Other cGHVD Sites**	ocular	10
	cutaneous	5
	pulmonary	2
	articular	3
	hepatic	3
	genital	1
	muscular	2
	none	7
**Average Age at Allogeneic-HSTC (years ± DS)**		51.1 ± 14
**Median Time between HSCT and Development of Oral GVHD (months)**		11

Table legends: cGVHD: chronic graft-versus-host disease; CCL: chronic lymphocytic leukemia; ALL: acute lymphoblastic leukemia; AML: acute myeloid leukemia; CML: chronic myeloid leukemia; MFI: idiopathic myelofibrosis; HL: Hodgkin’s lymphoma; NHL: non-Hodgkin’s lymphoma; PCL: plasma cell leukemia; MUD: Matched unrelated donor; ChT: chemotherapy; RT: radiotherapy; RIC: reduced intensity conditioning; MAC: myeloablative.

**Table 3 jpm-12-01628-t003:** Summary of oral and endoscopic features.

Patient	Localization	Oral Examination	Endoscopic Findings	Narrow Band Imaging	Histology
1	Buccal mucosa, gingiva, tongue body, bilateral labial commissure	Diffuse mucositis with erosive areas in the mucosa of the oral cavity. No erosive lesions of the tongue.	Diffuse mucosal erosive lesions without suspicion of malignancy.	No evidence of intrapapillary capillary loops	//
2	Buccal mucosa bilaterally and left tonsillar pillar with extension at amygdalo-glossus sulcus and oropharynx	Erythroplakia paired with central erosive area	Diffuse mucositis and evidence of erythroplakia with erosive central area suspicious for malignant disease	Evidence of intrapapillary capillary loops	Squamocellular carcinoma of the oropharynx.
3	Right buccal mucosa and lateral right surface of the tongue.	Leukoplakia of the right surface of the tongue with an indurated area in the lower portion.	Reticular lichenoid-like lesions on the right buccal mucosa without suspicious of malignancy	Evidence of intrapapillary capillary loops.	Epitelial verrucous hyperplasia
4	Bilateral buccal mucosa.	Reticular lesions with scleroatrophic changes of the mucosa.	Reticular lichenoid-like changes of the buccal mucosa bilaterally without suspicion of malignancy.	No evidence of intrapapillary capillary loops.	//
5	Bilateral buccal mucosa	Reticular lesions of the mucosa and evidence at buccal mucosa of the right side of small ulceration.	Lichenoid-like changes with an erosive area at the right buccal mucosa non-suspicious for malignancy	No evidence of intrapapillary capillary loops.	//
6	Oral vestibular mucosa, buccal mucosa, hard and soft palate.	Exophytic lesions with lichenoid-like changes on buccal mucosa and oral vestibular mucosa.	Lichenoid-like lesions on hard, soft palate and on oral pelvis without suspicious of malignancy	No evidence of intrapapillary capillary loops.	//
7	Bilateral buccal mucosa and tongue.	Ulcerative lesion on the left buccal mucosa with augmented peripheral vascularization. Erythematous appearance of the tongue with inflammation and edema of the lingual papillae.	Evidence of ulcerative lesion of the left buccal mucosa with erythroplasia-like changes and abnormal vascularization on its margins.	Evidence of dark spots on the border of the ulcerative lesion of the left buccal mucosa.	Delayed because of recurrence of intestinal adenocarcinoma.
8	Buccal mucosa, tongue, adherent gingiva, and hard palate.	Inflammation of the adherent gingiva with diffuse mucositis.	Erythematous lesions on tongue surface and buccal mucosa without suspicious of malignancy	No evidence of intrapapillary capillary loops.	//
9	Buccal mucosa and hard/soft palate	Blisters on hard and soft palate.	Lichenoid changes of the buccal mucosa bilaterally without suspicious of malignancy	No evidence of intrapapillary capillary loops.	//
10	Tongue, buccal mucosa, hard and soft palate.	Erosive lesions on the right buccal mucosa and on the left tongue lateral surface.	Diffuse erosive lesions on buccal mucosa bilaterally, lateral tongue surface, and lichenoid changes on the hard and soft palate. No suspicious of malignancy	No evidence of intrapapillary capillary loops.	//
11	Oral vestibular mucosa and buccal mucosa	Edema of the vestibular and buccal mucosa with florid sclero-atrophic lesions on cheek mucosa bilaterally	Diffuse lichenoid-like lesions of the oral cavity, no involvement of the tongue. No suspicious of malignancy	No evidence of intrapapillary capillary loops.	//
12	Oropharynx, buccal mucosa, and hard and soft palate	No evidence of lesions, diffuse hyperemia of the oral cavity	Diffuse mucositis of the oral cavity and oropharynx without suspicious of malignancy	No evidence of intrapapillary capillary loops	//
13	Oropharynx and buccal mucosa	Mucositis of the oral cavity and oropharynx with reticular lesions on buccal mucosa bilaterally.	Lichenoid-like chances of the mucosa without suspicion of malignancy.	No evidence of intrapapillary capillary loops	//
14	Buccal mucosa and hard palate	Diffuse mucositis of the oral cavity	Mucositis with hyperplasia of the buccal mucosa and hard palate bilaterally without suspicious of malignancy.	No evidence of intrapapillary capillary loops	//
15	Tongue and buccal mucosa	Glossitis and evidence of reticular lesions localized on buccal mucosa bilaterally	Lichenoid-like chances of the mucosa without suspicion of malignancy.	No evidence of intrapapillary capillary loops	//
16	Buccal mucosa and tongue base.	Ulcerative lesions localized at buccal mucosa bilaterally. Fungal infection at the tongue base.	No suspicious lesions.	No evidence of intrapapillary capillary loops	//
17	Buccal mucosa and oropharynx	Xerostomia and mucositis.	Enhanced vascularization pattern on buccal mucosa bilaterally without suspicious of malignancy	No evidence of intrapapillary capillary loops.	//
18	Buccal mucosa and hard palate	Erosive lesions on buccal mucosa bilaterally and hard palate	No suspicion of malignancy.	No evidence of intrapapillary capillary loops	//
19	Buccal mucosa and vestibular mucosa.	Diffuse blisters on the mucosa of the oral cavity and diffuse mucositis.	Demucosized area at left buccal mucosa without suspicious of malignancy.	No evidence of intrapapillary capillary loops	//
20	Buccal mucosa	Mucositis and lichenoid changing of the buccal mucosa bilaterally.	Bilateral erosive areas at cheek mucosa (major extension at left) without suspicious of malignancy.	No evidence of intrapapillary capillary loops	//

## Data Availability

The data that support the findings of this study are available on request from the corresponding author, C.P.

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
