# Peer review of "Implementation of Routine Endoscopy with Narrow Band Imaging in the Evaluation of Oral and Upper Airways Lesions in Oral Chronic Graft-Versus-Host Disease: A Preliminary Study"

_jpm, 2022, doi:10.3390/jpm12101628_

Round 1

Reviewer 1 Report

The abstract would be more clear if the authors mention the results findings of the patient with the suspicious lesion.

The manuscript would benefit from expanding on the role of narrow band imaging for oral lesion identification at the end of the introduction section rather than in the discussion section.

Table 2 would be better if changed to be a summary table for the diagnosis, type of donor, and conditioning rather than per individual data.

For the two patients with vascular abnormalities by NBI, what would be the clinical decision if NBI was not used, biopsy, monitoring, etc? Also, the course of these two patients would be better if described in the results section rather than in the discussion.

The last line on page 12 likely has a spelling mistake “the totality of patients was treated with a conditional chemotherapy regimen” Likely it is a conditioning chemotherapy regimen.

Author Response

Milan, September the 20th, 2022

Dear Editor,

We submit to your kind attention the revised version of our paper “Implementation of routine endoscopy with Narrow Band Imaging in the evaluation of oral and upper airways lesions in oral chronic Graft-Versus-Host-Disease: a preliminary study”. We’d like to thank you and the reviewers for our paper's accurate and positive evaluation. Following the editor’s and reviewers’ kind advice, we’ve performed the requested revisions. The performed revisions are not highlighted in the text as they would have hampered the readability of the content.

  • The abstract would be more clear if the authors mention the results findings of the patient with the suspicious lesion.
  • The abstract was modified in the “result” part to improve its efficacy as recommended by the reviewer.

  • The manuscript would benefit from expanding on the role of narrow band imaging for oral lesion identification at the end of the introduction section rather than in the discussion section.
  • Thank you for the option to clarify our message. We chose to add a full explanation of the role of NBI for oral lesions and oral GVHD in the introduction

  • Table 2 would be better if changed to be a summary table for the diagnosis, type of donor, and conditioning rather than per individual data.
  • Thank you for the suggestion, this has been modified in the manuscript.

  • For the two patients with vascular abnormalities by NBI, what would be the clinical decision if NBI was not used, biopsy, monitoring, etc? Also, the course of these two patients would be better if described in the results section rather than in the discussion.

  • As kindly asked by the reviewer we modified the discussion and result parts and we better described the clinical courses of our patients and the key role played by the use of NBI in the diagnostic process. In the manuscript we also make an attempt to describe the possibile scenario without the use of endoscopy with NBI.
  • The last line on page 12 likely has a spelling mistake “the totality of patients was treated with a conditional chemotherapy regimen” Likely it is a conditioning chemotherapy regimen.
    • The error has been amended, thank you!

Sincerely hoping that you shall find the revision complete and the article worthy of publication in your journal,

Looking forward to hearing from you

Kindest regards,

The authors

Reviewer 2 Report

The authors propose a retrospective study that aims to investigate the main oral lesion patterns in patients with oral Graft-Versus-Host-Disease and to describe and validate the use of endoscopy enhanced with narrow-band imaging (NBI) technique in the evaluation of the oral cavity and oropharynx for the early detection of oral premalignant disease and/or squamous cell carcinoma. 

It is an interesting study that fits the scope and profile of the journal however, several issues must be addressed. Please see the enclosed PDF

Author Response

Milan, September the 20th, 2022

Dear Editor,

We submit to your kind attention the revised version of our paper “Implementation of routine endoscopy with Narrow Band Imaging in the evaluation of oral and upper airways lesions in oral chronic Graft-Versus-Host-Disease: a preliminary study”. We’d like to thank you and the reviewers for our paper's accurate and positive evaluation. Following the editor’s and reviewers’ kind advice, we’ve performed the requested revisions. The performed revisions are not highlighted in the text as they would have hampered the readability of the content.

  • (Abstract Part) Results should be more elaborated on
  • The abstract was modified in the “result” part to improve its efficacy as recommended by the reviewers.

  • "A limit of our study is represented by the small sample analyzed. "This sentence can be eliminated from the abstract

  • Thank you for the suggestion, this has been modified in the manuscript.

  • The authors should discuss how age affects wound healing and disease prognosis, I suggest: Martu, M.A.; Maftei, G.A.; Luchian, I.; Popa, C.; Filioreanu, A.M.; Tatarciuc, D.; Nichitean, G.; Hurjui, L.-L.; Foia, L.-G. Wound healing of periodontal and oral tissues: Part II—Patho-phisiological conditions and metabolic diseases. Rom. J. Oral Rehabil. 2020, 12, 30–40.

  • Thank you for your suggestion. We underline the role of age in prognosis of oral lesions, indeed for oral GVHD. This has been added in the manuscript.

  • The discussions section is long and lacks focus. It should be better divided in sections, in different paragraphs.

  • We apologize! We make a major revision of the discussion part and we create different paragraphs (as recommended) to highlight our clinical message.

  • The authors should discuss potential differential diagnosis with other oral lesions, and amongst different age groups, I suggest: Popa, C.; Filioreanu, A.M.; Stelea, C.; Alexandru Maftei, G.A.; Popescu, E. Prevalence of oral lesions modulated by patients age: The young versus the elderly. Rom. J. Oral Rehabil. 2018, 10, 50–56.

  • We performed a review of the literature of oral potentially malignant disease and we added in the text the role played by age in oral lesions and in their prognosis. Thank you for the suggestion!

Sincerely hoping that you shall find the revision complete and the article worthy of publication in your journal,

Looking forward to hearing from you

Kindest regards,

The authors

Round 2

Reviewer 2 Report

The manuscript has been improved